# NADPH Oxidase 2 Mediates Myocardial Oxygen Wasting in Obesity

**DOI:** 10.3390/antiox9020171

**Published:** 2020-02-19

**Authors:** Anne D. Hafstad, Synne S. Hansen, Jim Lund, Celio X. C. Santos, Neoma T. Boardman, Ajay M. Shah, Ellen Aasum

**Affiliations:** 1Cardiovascular Research Group, Department of Medical Biology, Faculty of Health Sciences, UIT—The Arctic University of Norway, 9019 Tromsø, Norway; synne.s.hansen@uit.no (S.S.H.); jim.lund@ri.se (J.L.); neoma.boardman@uit.no (N.T.B.); ellen.aasum@uit.no (E.A.); 2School of Cardiovascular Medicine & Sciences, King’s College London British Heart Foundation Centre of Excellence, London SE5 9NU, UK; celioxcs@yahoo.com.br (C.X.C.S.); ajay.shah@kcl.ac.uk (A.M.S.)

**Keywords:** obesity, ROS, NADPH-oxidase, myocardial oxygen consumption, metabolism, cardiac efficiency

## Abstract

Obesity and diabetes are independent risk factors for cardiovascular diseases, and they are associated with the development of a specific cardiomyopathy with elevated myocardial oxygen consumption (MVO_2_) and impaired cardiac efficiency. Although the pathophysiology of this cardiomyopathy is multifactorial and complex, reactive oxygen species (ROS) may play an important role. One of the major ROS-generating enzymes in the cardiomyocytes is nicotinamide adenine dinucleotide phosphate (NADPH) oxidase 2 (NOX2), and many potential systemic activators of NOX2 are elevated in obesity and diabetes. We hypothesized that NOX2 activity would influence cardiac energetics and/or the progression of ventricular dysfunction following obesity. Myocardial ROS content and mechanoenergetics were measured in the hearts from diet-induced-obese wild type (DIO_WT_) and global NOK2 knock-out mice (DIO_KO_) and in diet-induced obese C57BL/6J mice given normal water (DIO) or water supplemented with the NOX2-inhibitor apocynin (DIO_APO_). Mitochondrial function and ROS production were also assessed in DIO and DIO_APO_ mice. This study demonstrated that ablation and pharmacological inhibition of NOX2 both improved mechanical efficiency and reduced MVO_2_ for non-mechanical cardiac work. Mitochondrial ROS production was also reduced following NOX2 inhibition, while cardiac mitochondrial function was not markedly altered by apocynin-treatment. Therefore, these results indicate a link between obesity-induced myocardial oxygen wasting, NOX2 activation, and mitochondrial ROS.

## 1. Introduction

Obesity, insulin resistance, and diabetes are independent risk factors for heart failure [1], and they are associated with a distinct diabetic cardiomyopathy independent of coronary heart disease or hypertension [2]. The pathophysiology behind this type of cardiomyopathy is far from elucidated, but increased oxidative stress is suggested to play an essential role [3,4]. A major non-mitochondrial source of ROS in cardiomyocytes are NADPH oxidases (NOXs). NOX2 is one of the main isoforms in the cardiomyocyte and several potent activators of NOX2, such as hyperglycemia, hyperlipidemia, angiotensin II, and cytokines [5,6], are known to be elevated in animals models of obesity and diabetes [7]. Experimental studies have demonstrated NOX2 upregulation in terms of gene expression, protein levels, and activity in the left ventricle of both type I [8,9,10,11] and type II [12,13,14] diabetic models. Pharmacological and genetic strategies for reducing NOX2 activity have also been associated with amelioration of oxidative stress and other factors that are believed to contribute to the development of diabetic cardiomyopathy, including fibrosis, ER stress markers, autophagy, and inflammation, in combination with improved ventricular function [7]. 

Important hallmarks of diabetic cardiomyopathy are altered myocardial metabolism with increased fatty acid oxidation [15,16,17], energy imbalance [15,18], impaired mechanoenergetic properties [19], and increased myocardial oxygen consumption (MVO_2_) [20]. The underlying mechanisms for the increase in MVO_2_ are not fully elucidated, but they may be linked to the increased fatty acid load [19] or oxidation [21], impaired calcium handing [22], and/or mitochondrial dysfunction [23,24].

Whether increased NOX2-mediated signalling might influence obesity/diabetes-induced impairments in myocardial metabolism and/or efficiency is not known. However, NOX2-signalling has been shown to influence many of the same factors that are believed to contribute to myocardial oxygen wasting in obesity/diabetes. These include changes in mitochondrial function and ROS production [25], as well as altered activity of calcium-handling proteins and impaired calcium homeostasis [26,27,28,29]. In accordance with this, we have also shown an association between cardiac tissue ROS-content and MVO_2_ in the hearts from diet-induced obese mice [23], suggesting a role for ROS in myocardial oxygen wasting. Based on this, we hypothesize that genetic ablation and the pharmacological inhibition of NOX2 may attenuate obesity-induced oxygen wasting in the myocardium.

## 2. Materials and Methods

### 2.1. Animals

Global NOX2 knock-out mice (NOX2 KO) and their wild-type littermates (WT) on a C57BL/6J background, which were originally obtained from a colony established at King’s College, London, UK, were bred locally. Five weeks old, male NOX2 KO mice (*n* = 16) and their wild-type male littermates (WT, *n* = 14) were fed a high a fat diet (HFD; 60% kcal from fat, TestDiet, London, UK) for 28 weeks, resulting in diet-induced obese mice (DIO_KO_ and DIO_WT_, respectively). Age matched WT (*n* = 7) and NOX2 KO (*n* = 7) mice that were fed a standard control diet served as lean controls (CON_WT_ and CON_KO_). 

In another cohort of animals, obesity was induced in male C57BL/6J mice (Charles River Laboratories, *n* = 42) by feeding them a Western, palatable diet for 18 or 28 weeks (WD, 35% kcal from fat, TestDiet, London UK) starting at the age of 5–6 weeks. After seven weeks on this diet, the mice were divided into two weight-matched groups receiving either normal water (DIO, *n* = 23) or water supplemented with the NOX2 inhibitor; 2.4 g/L apocynin (DIO_APO,_
*n* = 19 [30]) for the rest of the feeding periods. The age-matched mice were fed a standard control diet (CON, *n* = 22). Another subset of mice (DIO, DIO_APO_ and CON, 6–7 mice per group) were included to study mitochondrial function. 

The experiments were designed according to the guidelines from the Federation of European Laboratory Animal Science Associations (FELASA), EU animal research directive (86/609/EEC), Council of Europe (ETS 123), and the EU directive (2010/63/ EU). The local authority of the National Animal Research Authority in Norway approved the ethical protocols (FOTS id: 4772 and 3946). The 3R’s (Replacement, Reduction, and Refinement) have specifically been addressed when designing the study. All mice received chow *ad libitum*, free access to drinking water, and they were housed at 23 °C.

### 2.2. Plasma Parameters

To test glucose tolerance, blood was collected from the saphenous vein in fasted (4 h) animals and measured (glucometer, FreeStyle Lite, Alameda, CA) before (0 min.) and 15, 30, 60, 120 min. after administration of a glucose solution (1.3 g/kg body wt i.p.). Plasma free fatty acids (FFA) were analyzed using a commercial kit (NEFA-HR, Wako, Germany) and insulin was analyzed using ELISA kits (DRG Diagnostics, Germany). Insulin resistance was evaluated while using the homeostatic model assessment for quantifying insulin resistance (HOMA). This was calculated from the product of fasting blood glucose and insulin (μU/mL), and then divided by 22.5 [31].

### 2.3. Isolated Heart Perfusions 

The isolated hearts were perfused with a modified KHB buffer supplemented with 0.5 mM palmitate bound to 3% BSA. Myocardial glucose and fatty acid oxidation rates were measured in isolated perfused working hearts, while using radio-labelled isotopes [15]. Stroke work (SW) and parameters of left ventricular (LV) function were assessed by a 1.0 F micromanometer-conductance catheter. Myocardial oxygen consumption (MVO_2_) was measured using fiber-optic oxygen probes, as previously described [19]. The relationship between cardiac work and MVO_2_ was used to evaluate mechanical efficiency [19]. MVO_2_ was also measured in unloaded, retrograde-perfused hearts before (MVO_2 unloaded_) and after KCl-arrest, to measure oxygen cost for basal metabolism (MVO_2 BM_). Based on these measurements, we also calculated the oxygen consumption for processes related to excitation-contraction coupling (MVO_2 ECC_) [20].

### 2.4. Assessment of ROS Production in Myocardial Tissue

ROS production was assessed while using the dihydroethidium-high pressure liquid chromatography (DHE-HPLC) method, as described before [32]. Briefly, fresh LV tissue (58.7 ± 2.5 mg) was incubated with 100 µM DHE in PBS containing 100µM DTPA for 30 min. at 37 °C. The samples were quickly washed with PBS/DTPA, resuspended in cold acetonitrille and sonicated (3 × 8 W, 5 sec). The homogenates were centrifuged at 13,000 rpm for 10 min. and the supernatant collected in a new tube and dried under vacuum while using a SpeedVacuum (ThermoScientific). The dried pellet was frozen at −80 °C and further resuspended in PBS/DTPA and then injected into a HPLC system (Dionex, UltiMate 3000, London, UK). DHE was monitored by ultraviolet absorption at 245 nm and either hydroxyethidium (EOH) and ethidium (E) were monitored by fluorescence detection with excitation 510 nm and emission 595 nm. Here, the DHE-derived products are expressed as ratios of EOH and E per DHE consumed per gram of wet weight tissue. 

### 2.5. Mitochondrial Function and ROS Production

In a follow-up study, we sought to investigate the mitochondrial effects of NOX2 inhibition by apocynin treatment of C57BL/6J fed a WD for 28 weeks (CON, DIO, and DIO_APO_ mice). Isolated cardiac mitochondria were obtained from freshly harvested LV tissue while using a slight modification of the method of Palmer et al. [33], and mitochondrial function was measured in an oxygraph (Oxygraph 2k, Oroboros Instruments, Austria). The experiments were performed using malate (2.5 mM), glutamate (5mM) and pyruvate (10mM, PG), or malate (2.5 mM) and palmityol-carnitine (10mM, PC) as substrates. Mitochondrial leak state was determined before ADP addition (Leak). A saturated amount of ADP was then added to induce maximal mitochondrial respiration capacity (OXPHOS). Cytochrome C was added to the chambers (10 μM) to evaluate the integrity of mitochondrial membranes. H_2_O_2_ production was also assessed in respiring mitochondria while using Amplex UltraRed (AmR, 10 μM) dye. The reaction of H_2_O_2_ and AmR was catalyzed by horseradish peroxidase (10 uU) to produce a red fluorescent compound (recorded by the Fluo LED2-Module in Oxygraph 2k). The AUR signal was calibrated during the protocols using a fixed amount of H_2_O_2_ (10 μM). Exogenous SOD (5 μM) was added to reduce any remaining O_2_^−^ to H_2_O_2_. The rates of respiration and ROS production were adjusted to protein content and experiments were performed at 37 °C.

### 2.6. Real-Time Quantitative PCR

RNA was isolated from perirenal white adipose tissue (PWAT), stored in Allprotect Tissue Reagent, and then measured using the RNeasy Lipid Tissue Mini Kit protocol (Qiagen, Hilden, Germany). The liver samples were stored in RNA later (Qiagen, Hilden, Germany) at −20 °C until the isolation of RNA by RNeasy Fibrous Mini Kit protocol with a small modification (the Proteinase K step was removed). The RNA concentration and purity were determined using spectrophotometry (NanoDrop 2000, Thermo Fisher, Waltham, MA, USA). 200 ng (PWAT) or 1000 ng (liver) of RNA was reverse-transcribed into cDNA while using High Capacity cDNA Reverse Transcription Kit (Thermo Fisher, Waltham, MA, USA). Real-time PCR was performed in a LightCycler^®^96 System (Roche, Basel, Switzerland), where we pipetted 5 ng (PWAT) or 12.5 ng (liver) of cDNA and FastStart Essential DNA Green Master (Roche, Basel, Switzerland). Our target genes were Tumor necrosis factor-alpha (*Tnfα*) and NOX2 (*Nox2)*. *Tnfα* (NM_013693.3) forward primer: CAT-CTT-CTC-AAA-ATT-CGA-GTG-ACA-A and reversed primer: TGG-GAG-TAG-ACA-AGG-TAC-AAC-CC. *Nox2* (NM_007807.5) forward primer: TGAATGCCAGAGTCGGGATT and reversed primer: CCC-CCT-TCA-GGG-TTC-TTG-ATT-T. The target gene expression levels were normalized to hypoxanthine phosphoribosyltransferase 1 (*Hprt1,* NM_013556.2) detected by forward primer: TCC-TCC-TCA-GAC-CGC-TTT-T and reverse primer: CCT-GGT-TCA-TCA-TCG-CTA-ATC. The stability of the housekeeping gene was determined by geNorm [34].

### 2.7. Statistical Analysis

The data are expressed as mean ± SEM. Differences between groups were analysed using a One-Way ANOVA. A post-hoc test (Holm-Sidak method) was used with multiple comparisons between groups when using different genotypes (CON_WT_, CON_KO_, DIO_WT_ and DIO_KO_) and comparison against a control (DIO) when using the same genotype (CON, DIO and DIO_APO_). The overall significance level was set to *p* < 0.05. The differences between diets within the same genotype and differences between genotypes within the same diet are indicated within the tables and figures.

## 3. Results

### 3.1. Effect of NOX2 Ablation and Inhibition on Obesity, Glucose Tolerance and Inflammatory Status

Obesity promotes a low-grade chronic inflammation, which is associated with insulin resistance in mice. Therefore, we assessed both insulin resistance and a marker of hepatic and adipose tissue inflammation. Following 28 weeks on a HFD, DIO_WT_ and DIO_KO_ displayed similar gain in body-weight, PWAT, and liver weight when compared to their respective lean genotypes (CON_WT_ and CON_KO_, Table 1). An analysis of liver and white adipose tissue also showed increased hepatic and PWAT mRNA expression of NOX2 and the inflammatory marker TNFα (Table 1) in DIO_WT_ mice. Knocking down NOX2 did not reduce the hepatic expression of *tnfα* following obesity in DIO_KO_ compared to DIO_WT_ mice (Table 1). However, PWAT *tnfα* expression was significantly reduced in DIO_KO_ mice (Table 1), suggesting reduced adipose inflammation. The plasma levels of glucose and FFA were not different between groups, while fasted plasma insulin and HOMA-indexes were elevated in DIO_WT_ as compared to CON_WT_. Although there was also a trend towards reduced HOMA-IR in DIO_KO_ when compared to CON_KO_, this did not reach statistical significance.

Apocynin was added to the drinking water in order to reduce NOX2 activity in obese C57BL/6J mice to further study the effects of NOX2-activity in obesity. We used two time-points, 18 and 28 weeks of western diet (WD) feeding, in order to follow the development of obesity and systemic effects of NOX2 inhibition. In the first six weeks of apocynin treatment, body weight gain was significantly lower in DIO_APO_ when compared to untreated DIO mice (data not shown), but, at the 18 week time-point, there were no measurable effects of apocynin-treatment on obesity, liver, or PWAT weight (Table 2). However, there was an increased glucose tolerance and reduced levels of circulating FFA in DIO_APO_ mice as compared to non-treated DIO mice at this time point (Table 2). Weight gain increased to a similar extent in DIO and DIO_APO_ mice following 28 weeks of WD However, liver weights were reduced by apocynin-treatment (Table 2). Glucose tolerance was not attenuated by apocynin treatment in DIO_APO_ mice and, surprisingly, plasma FFA was not different between groups at this time-point (Table 2).

Together, these data suggest modest systemic effects of both NOX2 ablation and inhibition with a slight increase in glucose tolerance as well as an improved low-grade adipose inflammatory response to diet-induced obesity.

### 3.2. Effects of Obesity and NOX Inhibition on Myocardial Reactive Oxygen Species

Pieces of left ventricular (LV) tissue were incubated with DHE, and the fluorescence of the products hydroxyethidium (EOH) and ethidium (E) were measured using HPLC, in order to examine whether ablation or inhibition of NOX2 activity could reduce cardiac ROS-content. The EOH product is a specific superoxide-derived DHE product, and was increased in cardiac tissue from both models of obesity following 28 weeks of obesogenic diet when compared to their lean controls. The ablation and inhibition of NOX2 reduced the EOH product in cardiac tissue following obesity (Figure 1A,B). The ethidium product (which relates to other ROS) was not significantly elevated in ventricular tissue from obese mice or influenced by NOX2 ablation or inhibition (Figure 1C,D). 

### 3.3. Effects of NOX2 Ablation and Inhibition on Ventricular Function

Twenty-eight weeks of HFD induced LV dysfunction in DIO_WT_ mice. This was mainly a diastolic dysfunction, being illustrated by increased LV end-diastolic pressure (LVEDP) and end-diastolic pressure-volume relationships (EDPVR, Table 3). However, obesity-induced diastolic dysfunction was abrogated in DIO_KO_ mice, as these hearts exhibited reduced LVEDP and EDPVR when compared to DIO_WT_ (Table 3). In addition, the LV relaxation time-constant (Tau) was significantly lower in DIO_KO_ as compared to DIO_WT_ hearts, suggesting improved early LV diastolic function (Table 3). Although the parameters of LV systolic function (such as dP/dt max and Preload Recruitable Stroke Work index, PRSWi) were not significantly different between groups, we found cardiac output to be modestly increased in DIO_KO_ as compared to DIO_WT_ (Table 3).

Eighteen weeks of WD did not induce LV dysfunction in DIO mice, as the parameters of LV function were similar between groups (Table 4). Twenty-eight weeks of WD induced both LV systolic and diastolic dysfunction, being evident as lower cardiac output, reduced *dP/dt_min_*, increased relaxation factor (Tau) and EDPVR in DIO when compared to CON hearts (Table 4). Apocynin treatment increased LV systolic and diastolic function in DIO_APO_ mice, evident as increased cardiac output and PRSWi, together with the normalization of EDPVR relationships (Table 4).

Analysis of the relationship between LV end-diastolic pressure (LVEDP) and volume (LVEDV) at three different workloads showed a left and upwards shift of this relationship in DIO hearts following 28 weeks of obesogenic diet (Figure 2A,C), thus suggesting LV concentric remodeling. This was attenuated by both NOX2 ablation (DIO_KO,_
Figure 2A) and by apocynin treatment (DIO_APO_, Figure 2C). Ventricular volumes and pressures were not different between groups following 18 weeks of WD (Figure 2B).

### 3.4. Effects of NOX2 Ablation and Inhibition on Myocardial Energetics

Reduced LV mechanical efficiency (stroke work/MVO_2_) in hearts following obesogenic diets has been reported in several studies [23,34]. Here, we observed a trend towards reduced mechanical efficiency in DIO hearts following 18 weeks of WD (Figure 3B), with a significant reduction of LV mechanical efficiency in DIO when compared to CON hearts following 28 weeks of WD (Figure 3C). Ablation and inhibition of NOX2 both significantly increased mechanical efficiency in the hearts from obese mice (Figure 3A–C). The differences in LV mechanical efficiency were due to differences in MVO_2_ at the 18-week timepoint (Figure 3E), while both reduced MVO_2_ and increased SW seemed to contribute to increased LV mechanical efficiency following NOX2 ablation and inhibition after 28 weeks of obesogenic diets.

Obesity and diabetes have previously also been associated with elevated myocardial oxygen costs for non-mechanical processes (MVO_2unloaded_) [23,35], which includes oxygen consuming processes that are associated with excitation-contraction coupling (ECC) and basal metabolism (BM) to maintain cellular homeostasis [20,23]. In line with the changes in mechanical efficiency, DIO_KO_ hearts also exhibited reduced MVO_2unloaded_ compared to DIO_WT_ hearts when completely unloaded of mechanical work (Figure 4A). This was also found in DIO_APO_ hearts when compared to DIO hearts at both time points (Figure 4B,C). The reduced MVO_2unloaded_ in DIO_KO_ hearts was associated with reduced myocardial oxygen consumption for ECC (MVO_2ECC_, Figure 4D), and also found in DIO_APO_ after 18 weeks of obesogenic diet (Figure 4E). There were only subtle changes in the MVO_2_ for basal metabolism between groups (MVO_2BM_, Figure 4G,H). At the 28-week time point, both changes in MVO_2ECC_ and MVO_2BM_ (Figure 4F,I) seemed to contribute to a reduced obesity-induced oxygen wasting in unloaded hearts from DIO_APO_ mice.

### 3.5. Effects of NOX2 Inhibition on Myocardial Substrate Utilization and Mitochondrial Function

Myocardial glucose oxidation rates were markedly down in DIO as compared to CON hearts following 18 weeks of WD with a concomitant increase in palmitate oxidation rates. We did not find apocynin treatment to alter substrate oxidation rates (Figure 5).

We further studied the mitochondrial function in ventricular tissue from CON, DIO, and DIO_APO_ mice that were fed a WD for 28 weeks. The respiration rates were measured while using pyruvate and glutamate (PG) or palmityol-carnitine (PC) as substrates. There were no significant changes in Leak or OXPHOS states when using PG or PC as substrates in isolated cardiac mitochondria (Figure 6A,B,D–E). Respiratory control ratios (OXPHOS/Leak) were not significantly different between groups when using PG as substrates (Figure 6C), but significantly higher in the mitochondria from DIO_APO_ hearts when using PC as substrates. There was also with a borderline difference between CON and DIO (*p* = 0.08, Figure 6F). 

Finally, we assessed ROS production in isolated cardiac mitochondria. When adjusting the ROS production for mass or O_2_ flux (respiration rate), we found increased ROS production in the leak state in mitochondria from DIO hearts when using PG and PC as substrates. This obesity-induced ROS production was attenuated following apocynin treatment (DIO_APO_, Figure 7A,B). Similar results were found for ROS production in the OXPHOS states. In general, the mass-specific ROS production tended to be higher in the leak states than in the OXPHOS state within groups, while mitochondria respiring on PG exhibited higher ROS production than when respiring on PC (*p* < 0.001).

## 4. Discussion

In the present study, we only found subtle systemic effects of NOX2-inibition and ablation (KO) on the development of diet-induced obesity. Reduced NOX2-activity tended to improve insulin resistance and transiently reduce plasma lipids. It had little effects in fat depots, but tended to reduce adipose tissue inflammation. However, reduced NOX2 activity did have a marked effect on the development of cardiac dysfunction and the obesity-induced impaired cardiac energetics. For the first time, we were able to demonstrate that the obesity-mediated increase in myocardial oxygen wasting was prevented in both NOX2-KO mice and by NOX2-inhibition with apocynin. We also demonstrated a possible cross-talk between NOX2-activation and mitochondrial ROS-production, as has been previously suggested.

Apocynin-treatment and NOX2-ablation have been associated with reduced obesity in several studies on mice while using different ages, types of obesogenic diets, and feeding periods [30,36,37,38]. Many studies have also suggested NOX2-derived oxidative stress to be involved in the progression of obesity and pre-diabetes, through impaired glucose tolerance [30,37,39,40], insulin resistance [39], dyslipidemia [37,38,40], and visceral adipose inflammation [30,36]. However, there are discrepancies regarding the systemic effects of NOX2 activity in obesity, with one study also reporting adverse systemic effects, such as hyperphagia, elevated obesity, hepatic steatosis, and inflammation, together with exacerbated insulin resistance in obese NOX2 KO mice [41]. The present study does not show major effects of reduced NOX2 activity on body weight gain while using long term obesogenic diets, but the data do suggest a transient reduction following the start of the apocynin-treatment. This is in line with previous studies using shorter feeding periods and similar apocynin-treatments of DIO mice [30,38]. Additionally, reduced NOX2 activity seemed to improve glucose tolerance, reduce hyperlipidemia, and reduce adipose inflammation following obesity to some degree, in accordance with previous studies [30,36]. We did not find NOX2 ablation to reduce the expression of these inflammatory markers, as previously reported, although obesity in the present study was also associated with increased hepatic *nox2* gene expression and elevated markers of macrophage infiltration and inflammation in liver [30,38].

Even though the systemic effects of reduced NOX2 activity in obesity were subtle in our hands, the cardiac effects were more profound. Diabetes and obesity have been shown to increase myocardial NOX2 activity with both increased expression and recruitment of catalytic subunits to the plasma membrane and elevated ROS production [8,9,13,14,42,43]. Our data support this by showing elevated levels of EOH in cardiac tissue in DIO mice, which is primarily a product of increased superoxide. Apocynin treatment and the ablation of NOX2 partly normalized superoxide derived EOH, but had no effect on ethidium, suggesting reduced NOX2 activity when compared to reduced NOX4 (which primarily produces H_2_O_2_). Our results are also in line with previous data showing reduced cardiac NOX2 activity in diabetes and high fat feeding of mice following both the genetic ablation of NOX2 [8,44] and pharmacological treatment with NOX2-inhibitors [8,11].

In line with previous studies from our group using similar mouse strains, the use of a western diet seemed to produce more adverse cardiac effects in terms of inducing LV systolic and diastolic dysfunction [23] when compared to the use of a high fat diet which primarily induced LV diastolic dysfunction [34]. NOX2 knockout and apocynin-treatment both abrogated the development of obesity-induced LV dysfunction. This was evident as improved parameters of diastolic and systolic function and reduced concentric remodeling. Therefore, our results are in accordance with many studies where genetic and pharmacological strategies to reduce NOX2 activity have been associated with attenuation of LV dysfunction, pathological remodeling, in models of diabetes [8,9,11,12], sepsis-induced cardiomyopathy [29], pressure overload [45], and in the aged heart [28].

The present study also confirmed an obesity-mediated impairment of myocardial energetics, as previously reported in similar DIO models [22,23,34]. In accordance with previous studies [23], oxygen wasting processes were induced in the myocardium well before the development of LV dysfunction, as 18 weeks of WD induced elevated MVO_2_ for non-mechanical work (MVO_2unladed_) and processes that are associated with excitation-contraction coupling (MVO_2ECC_) in the absence LV dysfunction. Although not significant, the MVO_2_ used for basal metabolism also tended to be increased, however we were not able to reproduce previous studies showing the same decrease in mechanical efficiency or oxygen wasting effects following HFD [23,34,46]. One could speculate that also lean mice on the standard chow are becoming somewhat obese, and that a shorter feeding period in this mouse strain could have produced different results, due to the extensive feeding period. 

More importantly, we found that the mechanical efficiency was increased, while MVO_2unloaded_ and MVO_2ECC_ were reduced, suggesting improved cardiac energetics following NOX2 inhibition and ablation. The exact mechanisms behind obesity-induced myocardial oxygen wasting are not clear, but the increase in MVO_2ECC_ suggests that impaired calcium handling is a candidate. Many calcium handling proteins are redox sensitive and their activity may consequently be altered by NOXs [47]. Impaired calcium handling is well documented in diabetic hearts and includes altered sarcoplasmic reticulum Ca^2+^-ATPase2 (SERCa2) activity [48], elevated intracellular Ca^2+^-levels, and increased ryanodine receptor 2 (RyR2)-leakage [49]. Recently, Joseph et al. [44] reported profound effects of a short-term saturated high fat diet (SHFD, 4 weeks) of mice on myocardial calcium handling before the development of both obesity and ventricular structural changes. While SHFD induced heart rhythm abnormalities in WT mice, this was absent in NOX2 KO hearts and following apocynin treatment. Additionally, the oxidation of the RYR2 promotes calcium leak and increased calcium sparks, which were also absent in NOX2 KO hearts from mice that were fed a SHFD [44]. Other studies have reported a negative impact of NOX2-derived superoxide following acute exposure to FA load [25] and in other types of heart failure [28]. Our data demonstrating both improved LV function and myocardial energetics may very well be a functional consequence of the improved calcium handling associated with reduced NOX2 activity reported in the studies above. Wall stress has also been suggested to determine myocardial oxygen consumption [50], and it is partly determined by LV pressure. However, concentric remodeling could also reduce wall stress due to decreased LV radius and increased wall thickness. Although the total LV wall stress was not addressed in the current study, one cannot exclude that attenuated LV remodeling with the reduced LV stiffness observed by NOX2 ablation and inhibition could have contributed to improved mechanical efficiency.

Diabetes and obesity are associated with increased myocardial FA oxidation rates, which is again suggested to contribute to an obligatory increase in myocardial oxygen consumption, as the oxidation of FAs requires more oxygen for the same amount of ATP produced, when compared to glucose. However, there is evidence that the increased O_2_ cost for FA oxidation is lacking, and the inhibition of myocardial FA oxidation does not abolish the increase in MVO_2_ when hearts are perfused with high FAs [51,52]. Although we also found that increased FA oxidation in DIO hearts is accompanied by increased MVO_2_, we did not find the oxygen sparing effect of apocynin treatment to be linked to altered myocardial FA oxidation. This again supports that FA oxidation per se has no major role in altered MVO_2_. 

Several studies have associated diabetes and obesity with cardiac mitochondrial dysfunction, although there is no complete consensus in the literature [23,24,34]. Elevated, circulating FFA has been suggested to contribute, as an acute FA load to isolated hearts and cardiomyocytes has been shown to impair mitochondrial function [24,25]. The observed transient reduction in circulating FFA following apocynin treatment could very well be a contributor to the observed attenuation of detrimental cardiac effects. Although NOX2 is situated at the sarcolemma, Joseph et al. [25] found that both the inhibition and ablation of NOX2 could attenuate lipid-load induced mitochondrial respiratory dysfunction and mitochondrial ROS-release in cardiomyocytes. Maximal respiratory capacity was not impaired following obesity in cardiac mitochondria in the present study, but apocynin treatment was found to increase mitochondrial coupling. Improved mitochondrial coupling can be beneficial in terms of reducing oxygen-wasting processes for basal metabolism, and, therefore, could contribute to the reduced MVO_2unloaded_ observed following NOX2 ablation and inhibition in the present study. Although apocynin has been reported to exhibit direct antioxidant properties in vascular systems [53], it is extensively used as an inhibitor of myocardial NOX2 activity in many studies. Our data also support reduced myocardial NOX2 activity to be associated with reduced cardiac mitochondrial ROS release [25,54]. The apocynin-mediated reduction in ROS could contribute to the improved mitochondrial coupling in mitochondria from these hearts, as ROS have been shown to activate mitochondrial uncoupling proteins [55]. 

## 5. Conclusions

In line with previous studies on other models of heart failure, this study demonstrates that ablation and inhibition of NOX2 both attenuated obesity-induced left ventricular remodeling and dysfunction. Reduced NOX2 activity was also associated with improved myocardial energetics that could be attributed to decreased myocardial oxygen consumption for non-mechanical work, including processes that are associated with excitation-contraction coupling. Myocardial substrate utilization and mitochondrial respiratory capacity was not profoundly affected by NOX2 inhibition, but obesity-induced mitochondrial ROS production was abrogated. 

## Figures and Tables

**Figure 1 antioxidants-09-00171-f001:**
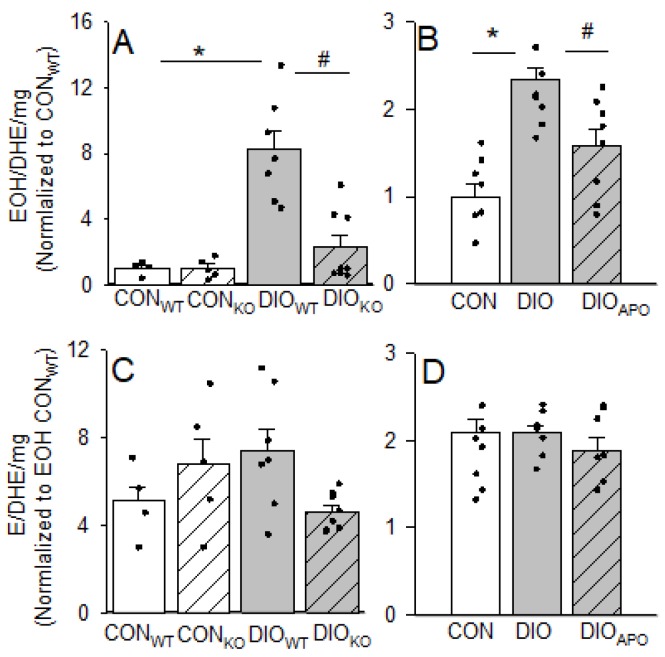
Reactive oxygen species-products hydroxyethidium (EOH) and ethidium (E) per dihydroethidium (DHE) consumed in cardiac tissue from lean controls (CON_WT_ and CON_KO_) and obese (DIO_WT_ and DIO_KO_) wild type and NOX2 KO mice fed a high fat diet for 28 weeks (**A**,**C**). Also shown in lean (CON), obese (DIO), and apocynin-treated obese (DIO_APO_) C57BL/6J mice fed a western diet for 28 weeks (**B**,**D**) *n* = 4–8 in each group. Values are normalized to EOH in lean controls. Single values and means ± SEM. * *p* < 0.05 CON vs. DIO within same genotype, ^#^
*p* < 0.05 DIO_WT_ vs. DIO_KO_ and DIO vs. DIO_APO_.

**Figure 2 antioxidants-09-00171-f002:**
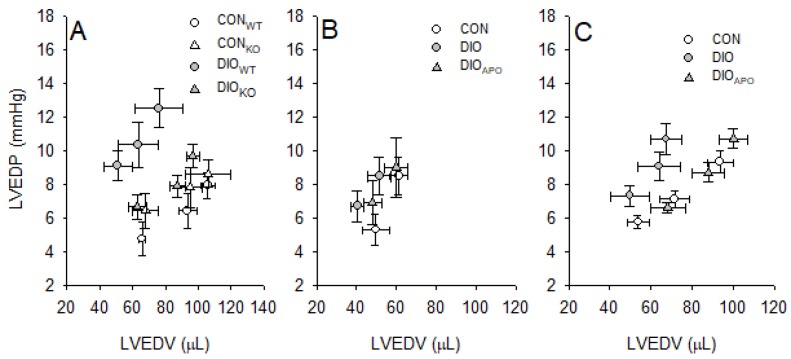
Steady state left ventricular end-diastolic volumes (LVEDV) and pressures (LVEDP) at three different workloads (preload: 4,6 and 8 mmHg and afterload: 50mmHg) from lean controls (CON_WT_ and CON_KO_) and obese (DIO_WT_ and DIO_KO_) wild type and NOX2 KO mice fed a high fat diet for 28 weeks (**A**), as well as lean (CON), obese (DIO) and apocynin-treated obese (DIO_APO_) C57BL/6J mice fed a western diet for 18 (**B**) and 28 weeks (**C**). *n* = 5–10 per group, the values are mean ± SEM.

**Figure 3 antioxidants-09-00171-f003:**
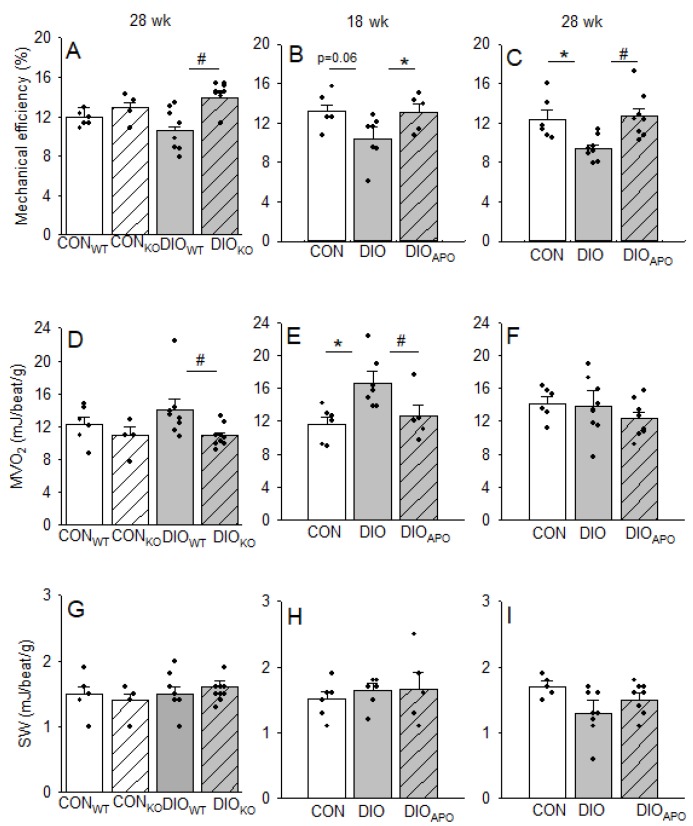
Left ventricular mechanical efficiency (**A**–**C**) expressed as stroke-work (SW, **G**–**I**) relative to myocardial oxygen consumption (**D**–**F**) in hearts from lean controls (CON_WT_ and CON_KO_) and diet-induced obese (DIO) wild type and NOX2 KO mice (DIO_WT_ and DIO_KO_) fed a high fat diet for 28 weeks (**A**,**D**,**G**) as well as lean controls (CON) and untreated and apocynin-treated obese C57BL/6J mice (DIO and DIO_APO_) fed a western diet for 18 weeks (**B**,**E**,**H**) or 28 weeks (**C**,**F**,**I**). Single values and means ± SEM. * *p* < 0.05 CON vs. DIO within same genotype, ^#^
*p* < 0.05 DIO_WT_ vs. DIO_KO_ and DIO vs. DIO_APO_.

**Figure 4 antioxidants-09-00171-f004:**
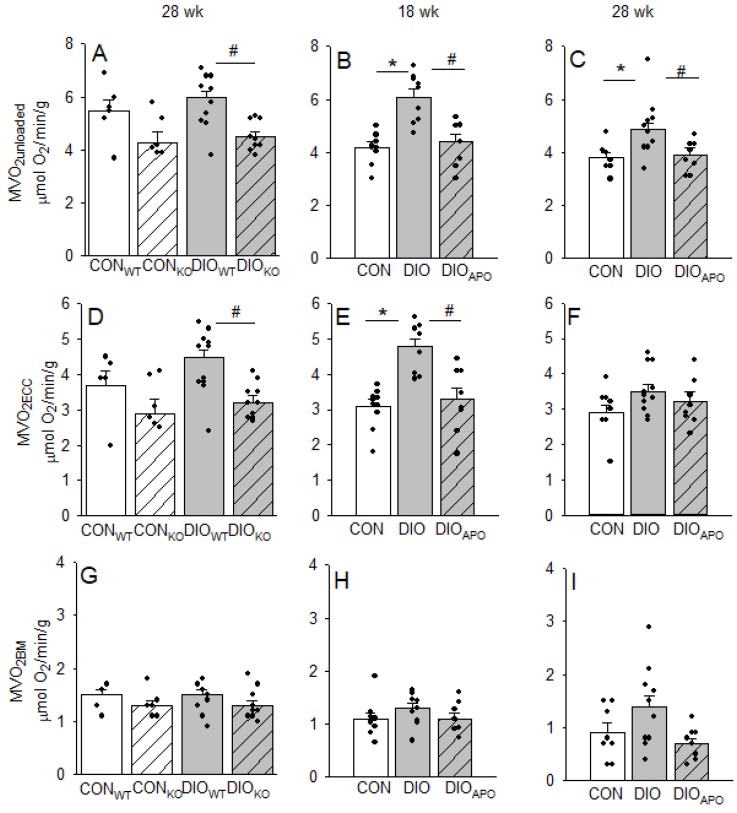
Myocardial oxygen consumption in mechanically unloaded hearts (MVO_2unloaded_, **A**–**C**) and for processes associated with excitation-contraction coupling (MVO_2ECC_, **D**–**F**) and basal metabolism (MVO_2BM_, **G**–**I**) that was obtained from lean controls wild-type (CON_WT_) and NOX2-KO mice (CON_KO_) and obese mice fed a high fat diet for 28 weeks (DIO_WT_ and DIO_KO_). (**A**,**D**,**G**). Also shown in lean controls (CON) and untreated and apocynin-treated obese C57BL/6J mice (DIO and DIO_APO_) that were fed a western diet for 18 weeks (**B**,**E**,**H**) or 28 weeks (**C**,**F**,**I**). Single values and means ± SEM. * *p* < 0.05 within the same genotype, ^#^
*p* < 0.05 between genotypes within the same diet.

**Figure 5 antioxidants-09-00171-f005:**
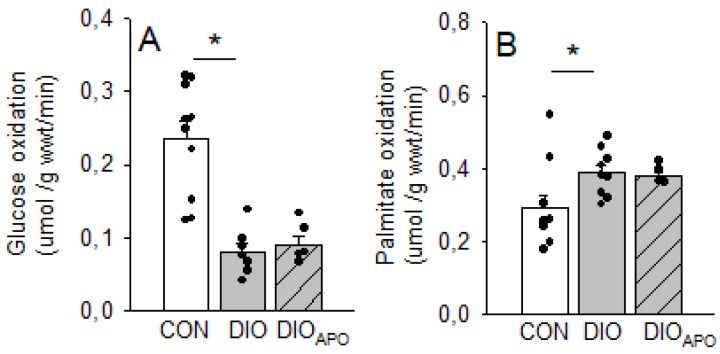
Glucose (**A**) and palmitate (**B**) oxidation rates measured in isolated working hearts from lean controls (CON), diet-induced obese (DIO), and obese apocynin-treated (DIO_APO_) C57BL/6J mice fed an obesogenic western diet for 18 weeks. Single values and mean ± SEM, * *p* < 0.05 vs. DIO.

**Figure 6 antioxidants-09-00171-f006:**
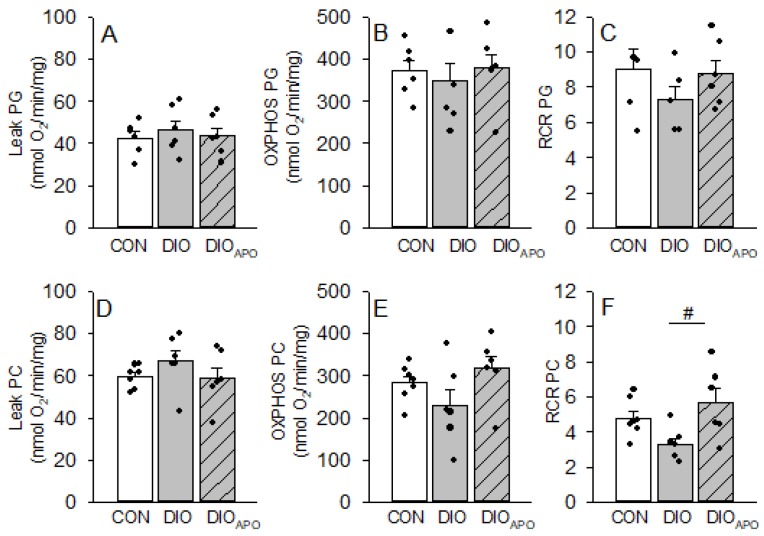
Oxygen fluxes (leak and maximal mitochondrial respiration capacity, OXPHOS) and respiratory coupling ratio (RCR) in isolated cardiac mitochondria using pyruvate and glutamate (PG, **A**–**C**) or palmityol-carnitine as substrates (PC, **D**–**F**). Mitochondria were obtained from lean controls (CON), untreated and apocynin-treated obese C57BL/6J mice (DIO and DIO_APO_) fed a western diet for 28 weeks. Single values and mean ± SEM, ^#^
*p* < 0.05 DIO vs. DIO_APO_.

**Figure 7 antioxidants-09-00171-f007:**
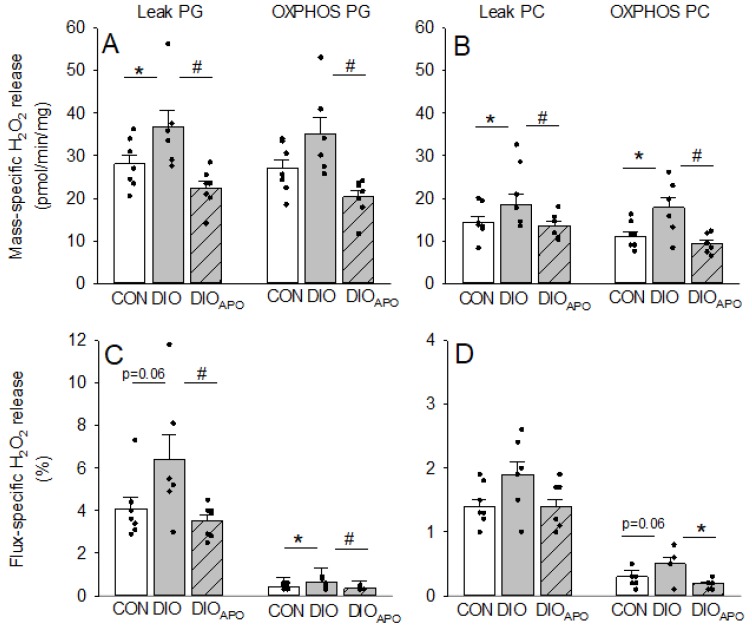
Mass-specific and flux-specific H_2_O_2_- release from isolated cardiac mitochondria using either pyruvate and glutamate (PG, panel **A** and **C**) or palmityol-carnitine as substrates (PC, panel **B** and **D**) as substrates. Mitochondria were obtained from lean controls (CON) and untreated and apocynin-treated obese C57BL/6J mice (DIO and DIO_APO_) fed a western diet for 28 weeks. Single values and means ± SEM, * *p* < 0.05 CON versus DIO, ^#^
*p* < 0.05 DIO vs. DIO_APO_.

**Table 1 antioxidants-09-00171-t001:** Animal characteristics of wild type (WT) and global NOX2 knock-out (KO), control mice (CON), and diet-induced obese mice (DIO) given an obesogenic diet for 28 weeks.

	CON_WT_ *n* = 7–9	CON_KO_ *n* = 7	DIO_WT_ *n* = 10–15	DIO_KO_ *n* = 10–16
Body weight (g)	35.1 ± 1.8	35.0 ± 1.0	50.8 ± 1.2 *	49.4 ±1.4 *
Heart weight (mg)	158 ± 3	159± 4	177 ± 6 *	183 ± 5 *
PWAT weight (g)	0.70 ± 0.12	0.61 ± 0.04	0.21 ± 15 *	0.22 ± 0.19 *
*Nox2* _PWAT_	1.0 ± 0.2	n.d.	5.0 ± 1.2 *	n.d.
*Tnfα* _PWAT_	1.0 ± 0.2	0.7 ± 0.1	10.2 ± 2.6 *	3.0 ± 0.6 *^,#^
Liver weight (g)	1.7 ± 0.2	1.5 ± 0.1	2.7 ± 0.2 *	2.6 ± 0.2 *
*Nox2* _liver_	1.0 ± 0.2	n.d.	1.9 ± 0.3 *	n.d.
*Tnfα* l_iver_	1.0 ± 0.3	0.9 ± 0.2	2.9 ± 0.4 *	2.8 ± 0.3*
Plasma FFA _fed_ (μM)	532 ± 112	967 ± 292	835 ± 96	1057 ± 260
Blood glucose _fasted_ (mmol/L)	5.6 ± 0.2	7.0 ± 0.1	6.6 ± 0.5	7.1 ± 0.3
Insulin _fasted_ (μU/mL)	1.7± 0.3	2.1 ± 0.4	6.5 ± 1.5 *	4.5 ± 0.5
HOMA-IR	2 ± 1	3 ± 1	30 ± 13 *	13 ± 3

Blood samples were obtained from *n* = 5–10 per group. Perirenal white adipose tissue (PWAT), free fatty acids (FFA). The mRNA expression of genes encoding for tumor necrosis factor α (Tnfα) and NADPH oxidase 2 (NOX2), was normalized to the corresponding expression in CON_WT_. Values are means ± SEM. * *p* < 0.05 CON vs. DIO within the same genotype. ^#^
*p* < 0.05 between genotypes in the same diet.

**Table 2 antioxidants-09-00171-t002:** Animal characteristics of chow fed controls (CON) and diet-induced obese C57BL/6J mice with or without apocynin treatment (DIO and DIO_APO_).

**18 Weeks of Diet**	**CON** ***n* = 8–11**	**DIO** ***n* = 8–12**	**DIO_APO_** ***n* = 8–9**
Body weight (g)	30.1 ± 0.4	42.7 ± 1.2 *	40.1 ± 1.8
Heart weight (mg)	134 ± 3	144 ± 4 *	149 ± 6
PWAT weight (g)	0.2 ± 0.2	1.0 ± 0.7 *	0.8 ± 1.1
Liver weight (g)	1.1 ± 0.1	1.7 ± 0.2 *	1.4 ± 0.1
Plasma FFA _fed_ (μM)	638 ± 88	1086 ± 143 *	648 ± 88 ^#^
Blood glucose _fasted_ (mM)	4.7 ± 0.2	5.1 ± 0.2	4.7 ± 0.1
Glucose tolerance test (AUC)	603 ± 45	945 ± 57 *	761 ± 52 ^#^
**28 Weeks of Diet**	***n* = 8–10**	***n* = 8–14**	***n* = 8–10**
Body weight (g)	31.6 ± 0.8	49.9 ± 1.2 *	46.5 ± 1.6
Heart weight (mg)	139 ± 4	158 ± 3 *	154 ± 4
PWAT weight (g)	0.4 ± 0.7	1.4 ± 0.6 *	1.2 ± 83
Liver weight (g)	1.1 ± 0.1	2.4 ± 0.2 *	1.8 ± 0.2 ^#^
Plasma FFA_fed_ (μM)	583 ± 44	637 ± 74	655 ± 39
Blood glucose _fasted_ (mM)	5.6 ± 0.3	6.0 ± 0.3	5.8 ± 0.3
Glucose tolerance test (AUC)	747 ± 55	926 ± 50 *	861 ± 67

Blood samples were obtained from 16–22 mice per group (18wk) and 10–14 in per group (28wk). Perirenal white adipose tissue (PWAT), area under curve (AUC). The values are means ± SEM. * *p* < 0.05 CON vs. DIO, ^#^
*p* < 0.05 DIO vs. DIO_APO_.

**Table 3 antioxidants-09-00171-t003:** Steady state and load-independent parameters of left-ventricular (LV) function obtained in isolated perfused working hearts from wild type (WT) and global NOX2 knock out (KO) control mice (CON) and diet-induced obese mice (DIO) given an obesogenic diet for 28 weeks.

	CON_WT_ *n* = 6	CON_KO_ *n* = 5	DIO_WT_ *n* = 6–11	DIO_KO_ *n* = 8–10
Cardiac output (mL/min)	11.6 ± 0.6	11.8 ± 0.9	10.9 ± 0.4	12.2 ± 0.2#
Coronary flow (mL/min)	4.0 ± 0.3	3.5 ± 0.3	3.8 ± 0.3	3.7 ± 0.2
Heart rate (bpm)	418 ± 1	408 ± 10	419 ± 7	413 ± 6
*dP/dt*_max_ (mmHg/sec)	3515 ± 259	3093 ± 126	3785 ± 230	3806 ± 143
*dP/dt*_min_ (mmHg/sec)	−2982 ± 215	−2611 ± 106	−3134 ± 174	−3230 ± 110
Tau _Weiss_ (msec)	10.8 ± 0.3	11.7 ± 0.3	11.7 ± 0.8	10.4 ± 0.3 ^#^
LVEDP (mmHg)	7.9 ± 0.8	8.6 ± 0.6	12.5 ± 1.2 *	9.7 ± 0.7 ^#^
LVDP (mmHg)	59 ± 2	59 ± 1	65 ± 2	66 ± 2
LVEDV (μL)	105 ± 4	106 ± 13	76 ± 14	97 ± 4
EDPVR (mmHg/µL)	0.12 ± 0.01	0.14 ± 0.01	0.24 ± 0.04 *	0.11 ± 0.01 ^#^
PRSWi	34.0 ± 1.1	28.8 ± 2.1	30.1 ± 2.9	36.7 ± 2.4

Hearts were paced at 10% above intrinsic heart rate. Steady state parameters were obtained at a pre- and afterload of 8 and 50 mmHg, respectively. LV end-diastolic pressure (LVEDP), developed pressure (LVDP) and end-diastolic volume (LVEDV), maximum positive and negative first time derivative of LV pressure (*dP/dt_max_* and *dP/dt_min_*), LV relaxation time constant (Tau). Load-independent parameters; the slope of LV end-diastolic-pressure-volume relationships (EDPVR) and Preload Recruitable Stroke Work index (PRSWi) were obtained by a temporary preload reduction. Values are means ± SEM. * *p* < 0.05 within the same genotype. ^#^
*p* < 0.05 between genotypes in the same diet.

**Table 4 antioxidants-09-00171-t004:** Steady state and load-independent parameters of left-ventricular (LV) function obtained in isolated perfused working hearts from chow-fed controls (CON) and diet-induced obese C57BL/6J mice with or without apocynin treatment (DIO and DIO_APO_).

**18 Weeks of Diet**	**CON** ***n* = 8**	**DIO** ***n* = 6**	**DIO_APO_** ***n* = 6**
Aortic flow (mL/min)	10.8 ± 0.4	10.1 ± 0.6	10.7 ± 0.1
Coronary flow (mL/min)	3.0 ± 0.2	3.3 ± 0.2	3.3 ± 0.3
Heart rate (bpm)	396 ± 11	359 ± 17	399 ± 13
*dP/dt*_max_ (mmHg/sec)	3749 ± 138	3925 ± 65	3615 ± 159
*dP/dt*_min_ (mmHg/sec)	−2854 ± 105	−2961 ± 89	−2863 ± 131
Tau _Weiss_ (msec)	11.0 ± 0.6	11.0 ± 0.4	11.3 ± 0.5
LVEDP (mmHg)	7.8 ± 1.1	8.5 ±1.1	9.0 ± 1.8
LVDP (mmHg)	57 ± 2	59 ± 2	57 ± 3
LVEDV (μL)	72 ± 10	58 ± 13	55 ± 14
EDPVR (mmHg/µL)	0.16 ± 0.01	0.19 ± 0.01	0.19 ± 0.04
PRSWi	29.9 ± 1.8	24.4 ± 1.2	27.4 ± 3.3
**28 Weeks of Diet**	***n* = 7**	***n* = 9**	***n* = 8**
Aortic flow (mL/min)	12.5 ± 0.3	9.4 ± 0.9 *	12.1 ± 0.1 ^#^
Coronary flow (mL/min)	3.2 ± 0.2	3.5 ± 0.2	3.2 ± 0.3
Heart rate (bpm)	434 ± 11	404 ± 9	415 ± 0.8
*dP/dt*_max_ (mmHg/sec)	3968 ± 50	3697 ± 121 *	3935 ± 92
*dP/dt*_min_ (mmHg/sec)	−3142 ± 51	−2735 ± 116 *	−2948 ± 68
Tau _Weiss_ (msec)	10.0 ± 0.2	11.3 ± 0.4 *	10.8 ± 0.3
LVEDP (mmHg)	9.3 ± 0.6	10.7 ± 0.9	10.3 ± 0.6
LVDP (mmHg)	59 ± 1	56 ± 1	58 ± 1
LVEDV (μL)	94 ± 6	67 ± 8 *	87 ± 2 ^#^
EDPVR (mmHg/µL)	0.18 ± 0.01	0.30 ± 0.04 *	0.18 ± 0.02 ^#^
PRSWi	28.3 ± 2.7	23.7 ± 2.4	32.4 ± 1.5 ^#^

The hearts were paced at 10% above intrinsic heart rate. Steady state parameters were obtained at a pre- and afterload of 8 and 50 mmHg, respectively. LV end-diastolic pressure (LVEDP), developed pressure (LVDP) and end-diastolic volume (LVEDV), maximum positive and negative first time derivative of LV pressure (*dP/dt_max_* and *dP/dt_min_*), LV relaxation time constant (Tau). Load-independent parameters including the slope of LV end-diastolic-pressure-volume relationships (EDPVR) and Preload Recruitable Stroke Work index (PRSWi) were obtained by a temporary preload reduction. Values are means ± SEM. * *p* < 0.05 CON vs. DIO. ^#^
*p* < 0.05 DIO vs. DIO_APO_.

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
