# Peer review of "NADPH Oxidase 2 Mediates Myocardial Oxygen Wasting in Obesity"

_antioxidants, 2020, doi:10.3390/antiox9020171_

Round 1

Reviewer 1 Report

This manuscript presents a study of NADPH oxidase 2 mediates myocardial oxygen wasting in obesity. The results were interesting and technically well done and demonstrated that both ablation and pharmacological inhibition of NOX2 improved mechanical efficiency and reduced myocardial oxygen consumption for non-mechanical cardiac work.  The results are very important to understand the link between obesity-induced myocardial oxygen wasting, NOX2 activation and mitochondrial ROS. This manuscript is ready for submission.

Author Response

This manuscript presents a study of NADPH oxidase 2 mediates myocardial oxygen wasting in obesity. The results were interesting and technically well done and demonstrated that both ablation and pharmacological inhibition of NOX2 improved mechanical efficiency and reduced myocardial oxygen consumption for non-mechanical cardiac work.  The results are very important to understand the link between obesity-induced myocardial oxygen wasting, NOX2 activation and mitochondrial ROS. This manuscript is ready for submission.

Answer: We thank the reviewer for the nice feedback on our study. We have done a final spell check and made amendments in manuscript.

Reviewer 2 Report

This manuscript investigated whether NOX2 activity influences cardiac energetics and/or the progression of ventricular dysfunction following obesity in mice model. The present study showed that inhibition of NOX2 by NOX2-knockout or NOX2 inhibitor apocynin improved mechanical efficiency and reduced myocardial oxygen consumption for non-mechanical cardiac work. In addition, the authors showed that mitochondrial ROS production was reduced following NOX2 inhibition. These results highlight a link between obesity-induced myocardial oxygen wasting, NOX2 activation and mitochondrial ROS. I think that the topic of this study and the obtained results are interesting. On the whole, in my opinion, this manuscript is suitable for the publication in Antioxidants.

Minor comments:

1. Why did not the authors examine the glucose tolerance test of NOX2 knockout mice?

2. There are careless mistakes (Line 125 and Line 126). The authors should revise them.

Author Response

This manuscript investigated whether NOX2 activity influences cardiac energetics and/or the progression of ventricular dysfunction following obesity in mice model. The present study showed that inhibition of NOX2 by NOX2-knockout or NOX2 inhibitor apocynin improved mechanical efficiency and reduced myocardial oxygen consumption for non-mechanical cardiac work. In addition, the authors showed that mitochondrial ROS production was reduced following NOX2 inhibition. These results highlight a link between obesity-induced myocardial oxygen wasting, NOX2 activation and mitochondrial ROS. I think that the topic of this study and the obtained results are interesting. On the whole, in my opinion, this manuscript is suitable for the publication in Antioxidants.

Minor comments:

Why did not the authors examine the glucose tolerance test of NOX2 knockout mice? There are careless mistakes (Line 125 and Line 126). The authors should revise them.

Answer: We thank the reviewer for the very nice feedback and constructive criticism.

Reviewer 3 Report

Provide experiments to show mitochondrial ultrastructure. Measure Oxygen Consumption rate (Seahorse). A dynamic measurement of mitochondrial ROS (mitoSOX) should be shown. Provide cardiac histology. "LV developed pressure end-diastolic pressure"?

Author Response

Comments and Suggestions for Authors

Provide experiments to show mitochondrial ultrastructure. Measure Oxygen Consumption rate (Seahorse). A dynamic measurement of mitochondrial ROS (mitoSOX) should be shown. Provide cardiac histology. "LV developed pressure end-diastolic pressure"?

Answer: We thank the reviewer for the feedback and constructive criticism. We have done a final spell check and made amendments in manuscript.

1 and 4.

We agree that pictures of mitochondrial ultrastructure and cardiac histology would have added additional information related to the role of NOX. Unfortunately, we have used the tissue for other purposes. We hope to include cardiac histology in future studies aimed to elucidate structural remodeling following obesity and diabetes.

2.We agree that seahorse experiments could to some extent increase the understanding of mitochondrial glycolysis (acidification rates), but we feel that the high resolution oxygraph (Oroboros O2k) experiments performed have provided solid data regarding the mitochondrial function in this study.  It also allowed us to do simultaneous ROS measurements.

3. The amplex red experiments from the Oroboros data are dynamic measurements of mitochondrial ROS production and continuously measured in parallel with mitochondrial respiration. We do not see the use of mitoSOX would gain considerable new information regarding dynamic mitochondrial ROS producttion.

5.“LV developed pressure end-diastolic pressure" in the table-texts (table 3 and 4) has been amended to LV end-diastolic pressure.

Round 2

Reviewer 3 Report

The Authors did not adequately address the Reviewers'concerns.